# UAV-based sampling systems to analyse greenhouse gases and volatile organic compounds encompassing compound specific stable isotope analysis

Simon Leitner[1], Wendelin Feichtinger[2], Stefan Mayer[2], Florian Mayer[2], Dustin Krompetz[3], Rebecca Hood-Nowotny[1], and Andrea Watzinger[1]

[1]University of Natural Resources and Life Sciences, Vienna, Institute of Soil Research, Tulln, Austria
[2]Combinnotec GmbH, Alland, Austria
[3]M3 Consulting Group, LLC, DBA M3 Agriculture Technologies, Phoenix, Arizona, United States

*Correspondence to*: Simon Leitner (simon.leitner@boku.ac.at)

**Abstract.** The study herein reports on the development and testing of sampling systems (and subsequent analytical setups) that were deployed on an unmanned aerial vehicle (UAV) for the purpose of analysing greenhouse gases (GHGs) and volatile organic compounds (VOCs) in the lower atmospheric boundary layer. Two sampling devices, both of which can be mounted to an unmanned aerial vehicle (UAV) with a payload capability greater than one kg, were tested for respective sampling and analysis of specific GHGs (carbon dioxide ($CO_2$) and methane ($CH_4$)) and VOCs (chlorinated ethenes (CE)). The gas analyses included measurements of the molar amounts and the respective stable carbon isotope ratios.

In addition to compound calibration in the laboratory, the functionality of the samplers and the UAV-based sampling was tested in the field. Atmospheric air was either flushed through sorbent tubes for VOC sampling or collected and sampled in glass vials for GHG analysis.

The measurement setup for the sorbent tubes achieved analyte mass recovery rates of 63 % - 100 % (more favourable for lower chlorinated ethenes), when prepared from gaseous or liquid calibration standards, and reached a precision (2σ) better than 0.7 ‰ for $\delta^{13}C$ values in the range of 0.35 – 4.45 nmol. The UAV-equipped samplers were tested over two field sampling campaigns designed to (1) compare manual and UAV-collected samples taken up a vertical profile at a forest site and (2) identify potential emissions of $CO_2$, $CH_4$ or VOC from a former domestic waste dump. The precision of $CO_2$ measurements from whole air samples was ≤ 7.3 µmol mol$^{-1}$ and ≤ 0.3 ‰ for $\delta^{13}C$ values and ≤ 0.03 µmol mol$^{-1}$ and ≤ 0.2 ‰ for $CH_4$ working gas standards. The results of the whole air sample analyses for $CO_2$ and $CH_4$ were sufficiently accurate to detect and localize potential landfill gas emissions from a secured former domestic waste dump using level flight. Vertical $CO_2$ profiles from a forest location showed a causally comprehensive pattern in the molar ratios and stable carbon isotope ratios, but also the potential falsification of the positional accuracy of an UAV-assisted air sample due to the influence of the rotor downwash. The results demonstrate that the UAV sampling systems presented here represent a viable tool for atmospheric background monitoring, as well as for evaluating and identifying emission sources. By expanding the part of the lower atmosphere that can practicably sampled over horizontal and vertical axes, the presented UAV-capable sampling systems, which also allow for compound-specific stable isotope analysis (CSIA), may facilitate improved understanding of surface-atmosphere fluxes of trace gas.

## 1 Introduction

Recent technical developments, the accessibility and the low cost of small unmanned aerial vehicles (UAV) have opened up opportunities for expanded sampling of the lower troposphere. As a result of increased societal environmental awareness and policy making efforts (Sikora, 2021), there is a growing demand to enhance monitoring of atmospheric trace gases, such as greenhouse gases (GHG) or volatile organic compounds (VOC). Recent studies have indeed shown that the deployment of small UAVs to sample the atmosphere for trace gases is a legitimate approach (Aurell et al., 2017; Barbieri et al., 2019; Rohi et al., 2020). Such UAV systems can be deployed to take air samples for subsequent laboratory analysis, as for the system presented here (Leitner et

al., 2020), or can be combined with low-cost sensors for on-board measurement and monitoring (U.S. Environmental Protection Agency (EPA), 2014).

Unmanned aerial systems (UAS), also referred to as remotely piloted aircraft system (RPAS), have a maximum take-off weight (MTOW) of < 25 kg and a maximum payload of < 4.5 kg and are defined as small UAVs. The type with a rotary-wing platform is particularly suitable for use in confined spaces, as they take off vertically, hover and have a high manoeuvrability (Burgués and

Marco, 2020). When combined with on-board samplers/sensors, these features thus expand the part of lower troposphere that can be potentially sampled, and furthermore allow for sampling/measurement along vertical profiles in the lower boundary layer (< 350 m above ground level (Chang et al., 2018)), which would otherwise require building towers or using balloons. By expanding horizontal and vertical sampling of atmospheric trace gas mole fractions, UAV systems could contribute to improved monitoring of atmospheric background levels and air quality, as well as improved inverse modelling of net surface-atmosphere fluxes.

Furthermore, the higher spatial resolution of UAV systems could improve the evaluation of sources and sinks of trace gases, if the measurements of mole fractions include additional stable isotope analysis of the respective trace gas compounds (e.g. Bergmaschi and Harris, 1995; Keeling et al., 1979; Randazzo et al., 2020; Whiticar, 1999; Widory et al., 2012; Zazzeri et al., 2017). There is thus scope for UAV systems to contribute to improved monitoring of GHG and air pollutant emissions, which is of utmost importance, when dealing with mitigation measures (Crotwell and Steinbacher, 2018) or law enforcement.

It is of course important to point out that the potential utility of UAV systems with respect to atmospheric monitoring, depends on the sampling and measurement instruments that can be carried on-board. Sampling and analysing the atmosphere for the compound specific mole fraction of GHG or VOC can be accomplished using a broad range of sampling systems and instrumentation. Compound specific isotope analysis, mainly relies on the utilization of mass spectrometry and laser or infra-red spectroscopy (Brewer et al., 2019). Such instrumentation depends on contextual sample specifications, like sample volume, sample vessel

tightness and avoidance of sample gas impurities, which are necessary for the sampling and measurement at low natural abundance of rare isotopic species of GHG and VOC. For VOC, sampling efforts can be significantly eased using sorbent tubes rather than heavy-weight canisters or large volume sample bags (Woolfenden, 1997). However, sampling methodology depends heavily on the targeted measurement precision. Sample pre-requisites for GHG measurements are similar to those of VOC, often relying on large and heavy sample containers (International Atomic Energy Agency, 2002), although there are versatile approaches using small and

light sample vessels (e.g. Górka and Lewicka-Szczebak, 2013). Notwithstanding this, the availability and access to towers and buildings represents a logistical limitation on spatial sampling of the near-surface atmosphere (Djuricin et al., 2010; Pataki et al., 2006; Takahashi et al., 2002). Such infrastructure is rendered potentially redundant when using UAVs equipped with versatile sensors or sampling devices. To date, most UAV-based approaches have focused on mole fraction measurements of GHG (Barbieri et al., 2019; Burgués and Marco, 2020) and a few preliminary applications of UAVs to perform whole-air sampling of GHG and

VOC have been documented (Chang et al., 2016). However, at the time of writing, there has been no published example of a UAV system to analyse the atmospheric mole fractions as well as the isotopologues from small sample vessel samplers. To our knowledge, there are currently no UAV-equipped sampling systems allowing for the subsequent quantification and stable isotope analysis of multiple GHGs and VOCs.

This study documents the development and testing of practicable UAV-based sampling systems and analysis pipelines tailored to

the analytical requirements for measuring multiple atmospheric trace gas species and their isotopologues. The aim was to develop gas-sampling devices that could be mounted onto small UAVs to sample atmospheric GHG and VOC, as an alternative to high-cost state-of-the art approaches typically applied at in situ monitoring stations. For GHGs, the sampling and measurement system was evaluated with respect to measurements of carbon dioxide ($CO_2$) and methane ($CH_4$). For VOCs, the focus was on measurements of chlorinated ethenes (CE), specifically Tetrachloroethene (PCE), Trichloroethene (TCE), *cis*-Dichloroethene (cDCE) and *trans*-

Dichloroethene (tDCE), which are commonly found in urban and industrial areas (Ras-Mallorquí et al., 2007). CE were sampled using sorbent tubes, while glass vials were used for GHG sampling and analysis. The co-developed measurement system was coordinated in such a way that it meshed with the sample vessels and ensured a correspondingly high quality measurement. GHGs

can be analysed directly in the sampling vessels, which overcomes any potential issues of leakage or loss, when samples have to be transferred to measurement vessels, as is the case when using gasbags for example (Chang et al., 2016; Greatwood et al., 2017).

Moreover, the systems described herein ensured detection limits were achieved well below the current atmospheric background values of 413 $\mu mol\ mol^{-1}$ and 1889 $nmol\ mol^{-1}$ for $CO_2$ and $CH_4$ respectively (WMO - World Meteorological Organization, 2021) and allowed to obtain a reasonable recovery rate of CE in sorption tubes. Furthermore, both collection systems should allow for the measurement of compound specific stable isotope ratios. On-board measurement (e.g. Khan et al., 2012; Martinez et al., 2020; Rohi et al., 2020) has numerous advantages and can obtain similar precision in molar ratios when compared to laboratory analysis (Shaw

et al., 2021). Nevertheless, an analysis system separated from the on-board sampling device allows for the measurement of the compound specific isotope ratios of multiple species and can allow for longer operation times due to the lighter payload.

The presented sampling systems consisted of a small UAV that was equipped with two different gas samplers (whole-air samples and sorbent tubes). The sampling systems, together with processes for pre-sampling whole-air sample vessel conditioning and post-sampling laboratory analysis, were tested and evaluated over two field sampling campaigns. First, UAV-based sampling of ambient

$CO_2$ over a vertical profile was compared to manual sampling at a forest site. Second, spatially distributed air samples from a former domestic waste dump aimed to investigate potential local GHG and VOC emissions.

## 2 Material and Methods

### 2.1 Gas sampling with sorbent tubes

#### 2.1.1 Sorbent tube preparation

Stainless steel tubes with an outer diameter of 6 mm and a length of 70.4 mm were used as sorbent tubes. They were filled from the bottom with sorbent material over a maximum length of 56 mm (equivalent volume of 1 mL), which was held in place by two pieces of 70 $\mu m$ mesh stainless steel gauzes. The sorbent tubes were filled with a gas stream top-down during sampling and bottom-up during desorption. A number of sorbent materials were tested, these were Carboxen 1016 60/80 mesh (Merck KGaA, Darmstadt, Germany), Molecular Sieve 5 Å 60/80 mesh (Sigma Aldrich Co., MO, USA), Porapak N 50/80 mesh (Waters Corporation, MA,

USA), HayeSep D 60/80 mesh (Hayes Separation Inc., Texas, USA) and Tenax GR 60/80 mesh (OHIO Valley Speciality, Ohio, USA). The adequate CE mass recovery potential (Brown and Shirey, 2001; Brown and Purnell, 1979; Ras-Mallorquí et al., 2007), the issue of CE-interfering ghost peaks (as also reported at Restek Corporation (2003)) appearing in blank-chromatograms and a maximum desorption temperature of 280°C of the available autosampler (VSP4000, Envea GmbH, Vohenstrauß, Germany) were decisive factors in choosing Tenax GR for the presented study. Although Tenax GR is a weak VOC-adsorbent it can therefore be

desorbed at much lower temperatures when compared to very strongly sorbent materials such as Carboxene 1016 (Ras et al., 2009). Prior to gas sampling, the sorbent tubes were conditioned at 300°C for 3 hours under a 20 $mL\ min^{-1}$ stream of nitrogen (purity 5.0) and then stored in glass tubes sealed with PTFE-caps. Tube conditioning was conducted using the TubeCon2 device (Envea GmbH, Vohenstrauß, Germany, Figure 1 (D)), which is a supplementary device to the purge and trap autosampler (VSP4000, Envea GmbH, Vohenstrauß, Germany) and was further discussed later-on in the text.

#### 2.1.2 Preparation of calibration standards.

The TubeCon2 device (D in Figure 1) was subsequently used to load sorption tubes with either liquid or gaseous calibration standard aliquots. Therefore, the TubeCon2 device was altered by using 15 cm long stainless steel pipes, bent upwards by 60° and substituting the sorbent tubes on the heating block (maintained at 80°C). The sorbent tubes were connected to the top of the bent pipes at the opposite end of a T connector sealed with 6 mm PEEK-fittings. The perpendicular end of the T, facing downwards at an angle of

30°, was used as an injection port for liquid or gaseous calibration standard aliquots and was sealed with a 3 mm PTFE coated silicon septa. Injected calibration standard aliquots were directed through the sorbent tubes providing a continuous nitrogen flow

(purity 5.0) of 20 mL min$^{-1}$ for 25 minutes, similar to previously described approaches (Hartwig, 2017; U.S. Environmental Protection Agency (EPA), 1999b). Gaseous aliquots were injected using a 100 µL gastight microliter syringe with a G26 side-port needle (SGE, BGB Analytik Vertrieb GmbH, Lörrach, Germany) and liquid aliquots were injected using a 10 µL gastight microliter syringe with a G26 bevel tip needle (SGE).

Calibrated compounds comprised of tDCE, cDCE, TCE and PCE (as pure substances, Sigma-Aldrich Handels Gmbh, Vienna, Austria). According to Woolfenden (1997) the use of organic solvents such as methanol is not suitable for Tenax GR, because methanol would also be retained causing instrumental issues at loading, desorption and analysis. The liquid calibration standard was prepared by injecting pure compounds into a 65 mL glass vessel filled with Millipore water, sealed with a Mininert cap (VICI AG, Schenkon, Switzerland) and stored at 5 °C (to give a final molar ratio of 400 µmol L$^{-1}$ each). Sorbent tubes were loaded with liquid standard aliquots of 1-10 µL (10-110 ngC), which was also applied in recent studies (Woolfenden, 1997).

The gaseous calibration standard was prepared by injecting liquid aliquots of tDCE, cDCE, TCE and PCE to a molar ratio of 1000 µmol mol$^{-1}$ made up in a 350 mL gas mouse (sealed with PTFE valves and a PTFE coated silicon septa), which had been equilibrated at 60°C for 30 min after flushing with Helium (purity 5.0) (U.S. Environmental Protection Agency (EPA), 1999b). After an initial equilibration time of 30 min a gaseous calibration standard was used over a period of two days, when stored at 60 °C. Gaseous aliquots of 10 to 100 µL (10-110 ngC) were transferred to the TubeCon2 device at a gas mouse and syringe temperature of 60 °C. The calibration range was designed to the manufacturer's (Thermo Fisher Scientific, Bremen, Germany) recommendation of linear IRMS analyser readings of 2 – 8 Volts, but could be further adjusted using the sample split option of the used purge and trap autosampler.

### 2.1.3    Gas sampler using sorbent tubes

The sorbent tube gas sampler (C in Figure 1) was loaded with four sorbent tubes (weight: 1200 grams; dimensions: 180x155x130 mm (LxWxH) with installed sorbent tubes). The sample gas inlet was set with a manually adjustable pinch valve, and maintained at 50 mL min$^{-1}$ when using 6 mm thick Tenax GR packed sorbent tubes (Brown and Purnell, 1979). All tubing was made out of 4/6 mm PTFE-tubing and tube connections were made out of polyethylene terephthalate (PET) or metal.

Passing the restriction valve at the sampler inlet the gas flow was split in two using a Y-connector and forwarded to the two inlet ports of an electric four-port gas distribution manifold. The gas manifold enabled switching between different sampling modes, either loading all four tubes simultaneously, individually or allowed for collection of subsequent duplicates. The sorbent tubes were installed at the outlets of the gas manifold using straight push-in connections. At the outlet of the four sorbent tubes the gas flow was merged into two streams using a 90° push-in Y-connector. Each stream then passed through a flow sensor, recording the actual flow rate. The gas streams were finally united and directed to the suction pump. To circumvent the non-regulated suction pressure of the pump a tee piece was installed prior to the pump feed to equalize the different flow rates set at the restriction valve of the gas inlet. Thereby the exposure to pressure differentials resulting in altered flow readings, the development of leaks and the overuse of sampler components could be prevented.

The sorbent tube sampler was equipped with an SD card, which besides recording the actual flow rate, collected and logged temperature, air pressure, humidity, the activated sample port number and time over the sampling event. The sorbent tube sampler was connected via a quick release dovetail mount to the bottom of the UAV.

## 2.2    Gas sampling with glass vials

### 2.2.1    Vial preparation and conditioning

Twenty mL crimp-top glass vials were used as sample vessels, which were sealed with 5 mm thick PTFE-lined grey butyl-rubber stoppers and aluminium crimp caps. The developed vial conditioning device (A in Figure 1) can be loaded with up to 12 glass vials

and conditioned the vials via flushing and evacuation. First, crimp sealed vials were flushed with synthetic air or helium for 1 min at 200 mL min$^{-1}$ using two G26 side-port needles. Second, flushed vials were evacuated through a single G26 side-port needle to a final pressure of approx. 0.5 Pa using a rotary vane pump (Edwards E2M-1.5, Sussex, UK). In order to follow for the identical-treatment principle (Werner et al., 2001), vials used for whole-air sampling and for preparing compound calibration standard vials were all flushed with synthetic air (e.g. preventing matrix effects during analysis).

### 2.2.2 Whole air sampler

The whole air sampler (B in Figure 1) weighed less than one kg (200x200x200 mm) and could be loaded with up to 12 glass vials positioned in a rotating barrel. The sample gas inlet was positioned at a vertical offset of 40 cm to the centre of the UAV rotor-plane in order to minimize the impact of the airflow from the UAV rotors (Alvarado et al., 2017; Palomaki et al., 2017; Poyi et al., 2016; Zhou et al., 2018). A 0.5 mm ID PEEK-tubing (length of approx. 70 cm) was used as transfer line to connect the downwards facing sample inlet at the centre of the UAV rotor-plane to the gas inlet of the whole air sampler, which was mounted to the bottom of the UAV platform. The gas inlet consisted of a G23 side-port needle (Hamilton Bonaduz AG) mounted to a moving cantilever. At a sampling event the cantilever pushes the needle through the glass vial septa and thereby enables the evacuated vial to equilibrate with the surrounding environment, sucking in a gas sample of approx. 20 mL (equilibration time of 25 sec). The dead-volume of the transfer line was 100 µL and the residual flush-gas volume inside the evacuated glass vial was < 0.5 mL. Analysis of evacuated glass vials, filled thereafter with synthetic air only, did not reveal any chromatographic peak for $CH_4$ and a reproducible blank peak of approx. 30 µmol mol$^{-1}$. The consistent blank peak was either due to the impurity of the flushing gas bottles ($CO_2 \leq 0.5$ µmol mol$^{-1}$, Synthetische Lut 5.0 KW-frei, Messer Austria GmbH, Gumpoldskirchen, Austria) or a leak somewhere in the buildings gas pipe system, because blank vials prepared and flushed with Helium (purity 5.0) did not show chromatographic peaks at the retention time of $CO_2$.

### 2.3 UAV description

The UAV used during the field sampling campaigns was a Hermes V2 RPAS (Figure 1, M3 Agriculture Technologies, Dayton, OH, USA), which is a 1000 mm (motor to opposing motor) scale hexacopter utilising an ArduPilot supported autopilot and associated hardware. The Hermes V2 is capable of operating in conditions such as high wind (< 20 knots) and light rain due to its design, which places sensitive electronic components inside a fuselage protected from rain.

ArduPilot is a community supported open source autopilot software suite supporting a variety of autonomous ground, water and air vehicles. The user interface or Ground Control Station (GCS) utilized to plan the sampling operations and interface with the Hermes V2 is MissionPlanner, an open source GSC software which supports ArduPilot. The Hermes V2 can lift up to five kilograms of payload and operate for up to 25 minutes when equipped with zero payload, while drawing energy from a 17,000 mAh 6S lithium polymer (LIPO) High Voltage battery. The time aloft of any RPAS (remotely piloted aircraft system) is inversely proportional to the mass of the payload. The Hermes V2 weighs 7.25 kg when ready to fly. The atmospheric samplers utilized during the sampling campaign each weighed less than 1.5 kg and allowed maximum flight times up to 22 minutes, depending upon environmental and mission planning requirements. Technical details of the Hermes V2 RPAS are provided in the supplementary material (Table S1). The gas samplers were triggered to gather a sample utilizing a 5 V DC relay connected to the open source autopilot. The relay was autonomously triggered with missions created using MissionPlanner GCS. Sample collection was initiated by approaching within 2 meters of a designated point, where the relay would be triggered. The sampling mission was programmed to delay and gather a sample at the designated point for 25 (to glass vessels) or 600 seconds (to sorption tubes). The RPAS could then move to another sampling location or return and land at the take-off location. A sample could alternatively be collected manually utilizing the pilots console transmitter. The samplers were mounted underneath the RPAS fuselage between the landing gear legs using a quick release dovetail mount. Twelve V DC power was supplied to the atmospheric samplers from the RPAS. Flight logs were available to be

downloaded from the autopilot and analysed post flight using the MissionPlanner GCS software. Flight profiles could be visually appreciated by viewing a .kmz file and other data such as sampling heights and GPS coordinate locations could be confirmed.

**Figure 1: Overview of the sampling systems comprising of the vial preparation device (A), the whole-air sampler (B), the gas sampler for adsorbent tubes (C), the sorbent tube conditioning device (TubeCon2, D) and the UAV applied in field test equipped with the whole-air sampler.**

## 2.4    Referencing and calibration of stable carbon isotope ratios

Stable isotope ratios of carbon in $CO_2$, $CH_4$, PCE, TCE, cDCE and tDCE were reported in the δ-notation (‰) and were referenced to the Vienna Pee Dee Belemnite (VPDP) scale. The normalisation of measured stable isotopic compositions to isotope reference scales followed the procedures of Paul and Skrzypek (2007). The δ-values were calculated as

$$\delta^{13}C \quad = \frac{R_P}{R_{Std}} - 1,$$

where R is the ratio of the abundance of $^{13}C$ to $^{12}C$ of a sample (P) and a measurement standard (Std) (Coplen, 2011). $\delta^{13}C$ values of CE were calibrated against three international reference materials (USGS 87, NBS 22, IAEA CH-3) using an elemental analyser connected to a DeltaV Advantage IRMS (EA-IRMS, Thermo Fisher Scientific, Bremen Germany). Assigned $\delta^{13}C$ values of CE were -27.51‰ ± 0.13‰ (n=5), -29.81‰ ± 0.08‰ (n=3), -25.94‰ ± 0.02‰ (n=5), -12.22‰ ± 0.02‰ (n=5) for PCE, TCE, cDCE and tDCE, respectively. Both, CE and reference materials, were sampled in tin cups designed for sampling liquids. The $CO_2$ and $CH_4$ working gas was calibrated against two isotope certified $CO_2$ gas standards (-6.7‰ ± 0.2‰, -39.0‰ ± 0.2‰, ISO-TOP, Messer Austria GmbH) after direct injection to a GC-C-IRMS measurement setup, as presented at Leitner et al. (2020). Obtained $\delta^{13}C$ values ± 1 σ were -4.34‰ ± 0.2‰ (n=17) and -40.3‰ ± 0.2‰ (n=38) for the $CO_2$ and $CH_4$ working gases respectively.

## 2.5    Measurement setup for sorbent tubes

The measurement system (purge&trap GC-qMS/C-IRMS) comprised a purge and trap autosampler (VSP4000, Envea GmbH, Vohenstrauß, Germany) connected to a gas chromatograph (GC, Trace GC, Thermo Scientific, Bologna, Italy) linked at a 10:1 gas flow split ratio to a gas conversion system (GC-Isolink, Thermo Fisher Scientific, Bremen, Germany) and a single-quadrupole mass spectrometer (qMS, ISQ, Thermo Fisher Scientific, MA, USA). The qMS was in electron ionization mode with the filament emission at 70 eV and a source temperature of 230 °C to detect the m/z ratios of mass 12 to 166 at a scan time rate of 0.2 sec. The GC-Isolink was further connected to a gas distribution system (Conflo IV, Thermo Fisher Scientific, Bremen, Germany) introducing the $CO_2$-converted gaseous analytes together with $CO_2$ working gas spikes to an isotope ratio mass spectrometer (IRMS, Delta V Advantage, Thermo Fisher Scientific, Bremen, Germany). The mass-to-charge rations (m/z) of mass 44, 45 and 46 were continuously monitored to quantify the amounts of each analyte and determine its stable carbon isotope ratio (as $\delta^{13}C$).

Sorbent tubes were analysed using the purge and trap autosampler set to thermal desorption mode. Thereafter, sorbent tubes were heated to 200 °C to desorb analytes during a period of 10 min and transferred with a helium flow of 20 mL min$^{-1}$. Adsorbed water vapour was removed by a membrane water trap (purged with $N_2$ at 200 mL min$^{-1}$). Desorbed analytes were trapped at -50 °C inside a Tenax GR-packed cryotrap cooled with liquid nitrogen ($LN_2$) and then released by heating the cryotrap to 200°C to be transferred with the He carrier flow (inlet pressure of 1200 mbar) to the GC, equipped with a 30m, 0.25 mm ID, 0.25 μm film thickness TG-5MS column (Thermo Fisher Scientific, Bremen, Germany). The temperature program started at an initial temperature of 35 °C, held for 1 min, then heating-up to 70 °C by 5 °C min$^{-1}$, before reaching the final temperature of 260 °C after heating-up by 60 °C min$^{-1}$.

## 2.6    Measurement setup for glass vials

The measurement procedure and the preparation of calibration standards for the analysis of $CO_2$ and $CH_4$ (molar ratio and $^{13}C/^{12}C$ ratio) can be found in detail in a preceding publication of Leitner et al. (2020). The analysis of $CO_2$ (detection limit of 100 μmol

mol$^{-1}$) was carried out with head-space (HS) GC-C-IRMS analysis. A 300 uL sample aliquot was injected via an autosampler (CTC Combi PAL, Switzerland) to a ShinCarbonST 80/100 mesh 2m x 1mm ID packed GC-Column (Restek Corporation, BGB Analytik AG, Rheinfelden, Switzerland). The temperature programme of the GC starting at 40°C, heated up by 20 °C min$^{-1}$ to 150 °C, held for 5 min, before heated up by 50 °C min$^{-1}$ to the final temperature of 180 °C. $CO_2$ was then passed through the non-active (400 °C) high-temperature-conversion unit inside the GC Isolink, to assure an unchanged state of $CO_2$ before being sent to the IRMS (Delta V Advantage).

The analysis of $CH_4$ for atmospheric background levels (~1.9 µmol mol$^{-1}$) was carried out using a different measurement setup. Analysis of $CH_4$ (detection limit of 0.7 µmol mol$^{-1}$) followed a purge and trap autosampling routine using a VSP4000, set to the purge and trap mode) equipped with a HayeSep-D (60/80 mesh) packed cryotrap maintained at -140 °C using $LN_2$ and subsequent cryogenic trapping at the initial section of a Poraplot Q (30 m, 0.32 mm ID) GC-column (Agilent Technologies Austria GmbH, Vienna, Austria), inside a $LN_2$ dewar, which is otherwise maintained at 35 °C inside the GC. $CH_4$, which was thereby separated from the interfering atmospheric air components (e.g. $N_2$, $CO_2$ and $N_2O$), was then oxidized to the measurement gas $CO_2$ by passing through a combustion/reduction reactor (GC Isolink) before being forwarded to a ConfloIV linked to a Delta V Advantage to measure the stable isotopic composition of carbon.

## 2.7    Description of field sites

The gas sampling system was tested at two field sites, which were representative for the application of the sorbent tube and/or whole air sampling system. Target compounds using the sorbent tube sampler were VOC such as CE, which are a prominent constituent of encapsulated and secured former domestic waste dumps across Europe. The whole air sampler was designed to specifically sample the atmosphere for greenhouse gases.

The former domestic waste dump at Kapellerfeld (Lower Austria, Austria, https://www.altlasten.gv.at/atlas/verzeichnis/Niederoesterreich/Niederoesterreich-N12.html), where CE and BTEX (benzene, toluene, ethylbenzene, xylenes) had been identified as part of the pool of potential local contaminants, was chosen for testing the sorbent tube sampler. Due to the formation of landfill gas at Kapellerfeld local authorities had installed a landfill gas extraction system to prevent emission, which mostly consist of $CH_4$ and $CO_2$. In order to test whether the sampling system was capable of detecting potential leakage through encapsulated landfills or piping systems the whole air sampler was also used at Kapellerfeld.

The whole air sampler was tested along a horizontal sampling profile above the interconnecting pipelines of two landfill gas suction system units. Each unit consisted of an above-ground pipeline with alternate gas extraction wells connected to the pipeline at right angles. The above-ground pipelines of both units were aligned to each other, but were also broken half-way in between at a perpendicular transfer pipeline. At the sampling event, only one unit was operating and there was a total of 23 gas extraction wells. The flight path started 3 m above the gas extraction wells on one side of the operating unit and continued until the unit in "stand-by" before reversing over the opposite sided extractions wells back to the starting point. Independent single samples were taken, which were analysed for the carbon isotope ratio and mole fraction of $CO_2$ and $CH_4$. Each compound was measured sequentially from the same sample vessel filled at the waste dump using two different measurement setups (Leitner et al., 2020). First, $CO_2$ was analysed from three measurements of 300 µL sample volume aliquots each, before analysing the entire residual volume (~19.6 mL) for $CH_4$.

The sorption tube sampler was tested on a horizontal and vertical sampling profile at the ex-situ filter facility of the former domestic waste dump. The filter facility surroundings had a noticeable odour that day. Sorbent tubes of the vertical sampling profile were flushed with ambient air at a flow rate of 50 mL min$^{-1}$ for an individual sampling time of 10 minutes. Discreet single samples were taken (sampling mode 4x1) at 7, 10 and 20 meters above ground level (the 4th sorption tube position was kept unloaded and used as a sample blank). The horizontal sampling flight took place over a covered observation well of the local funnel and gate system. An ambient air sample was taken at a fixed height of 3 m above the well, in quadruplicate (sampling mode 1x4), with a pumping rate of 200 mL min$^{-1}$.

The second field campaign took place at the forest demonstration centre of the University of Natural Resources and Life Sciences, Vienna, located in Forchtenstein (Burgenland, Austria), to assess the positional accuracy of UAV-based whole air sampling. For vertical $CO_2$ profiling, manual samples were collected at six heights during the ascent to an observation tower with a final sample after the descent at the height of 0.4 m, similar to the position of the first sample. Subsequently, sampling was conducted with the air sampler in the immediate vicinity of the tower at six comparable and two additional heights. Prior to drone launch, additional samples were collected with the rotors turned on using the air sampler mounted to the UAV, which was not in flight mode. In addition, before the rotors were turned on, simultaneous manual and UAV-assisted sampling was carried out. All samples were taken in triplicate, with the exception of the sample at the descent from the tower, which was taken as a single sample.

The 36 m high tower is located in a mixed forest with a canopy height of about 20 m. Since the tower exceeds the canopy height by 16 m, it should be possible to capture the atmospheric $CO_2$ background in addition to the area strongly influenced by the soil and vegetation. The field sampling campaign took place in October 2021 with overcast weather conditions during sampling and temperatures around 8°C. Generated vertical profiles of $CO_2$ were carried out to draw conclusions whether the UAV-based sampling system meets the requirements for investigating net fluxes and identifying sources and sinks.

## 3    Results and Discussion

### 3.1    Calibration of chlorinated ethenes using sorbent tubes

Two calibration standards, one prepared by diluting pure liquid phase CE in an aqueous phase and a second using vaporized CE in a gas phase (He) were used for the calibration of the thermal desorption (TD) method. Their results were evaluated based on mass and $\delta^{13}C$-value recovery. The liquid phase calibration standard was first measured against other CE-containing laboratory working standards to check for the accuracy of assigned set values. This was accomplished by measuring liquid standard aliquots with a purge and trap GC-C-IRMS measurement setup described in Leitner et al. (2018). As with the latter, the TD method development was carried out using the same GC-C-IRMS instrumentation to enable the comparison of peak areas in order to check for the mass recovery of CE when loaded to sorbent tubes. A comparison of peak areas obtained from both measurement setups showed that, according to a Student-t-Test (Student, 1908), peak areas per injected mass of CE obtained by TD were not significantly lower (Table S2). In addition, incomplete loading of the sorbent tubes (compound breakthrough) would lead to a significant depletion in compound's $\delta^{13}C$ values (Klisch et al., 2012). Liquid standard aliquots were calibrated over a range of 0.35 – 4.45 nmol on GC-column (corresponding IRMS mass 44 intensity range: 100 to 8000 mV). Set values of liquid standard aliquots showed a linear correlation with peak areas ($R^2 \geq 0.98$), valid for all CE, and on a 1:1 relationship.

**Table 1. Comparison of $\delta^{13}C$ mean values ± twice the standard deviation (2 σ) and mass recovery rates, as means ± 2 σ, obtained from measurements of gaseous and liquid calibration standard aliquots at the given mass range (nmol) of chlorinated ethenes (PCE, TCE, cDCE, tDCE) loaded to sorbent tubes.**

Tschickardt et al. (2017) recommended to calibrate TD-methods with test gases, spiking liquid stock solutions to a gas stream. The conducted approach of spiking gaseous calibration standard aliquots to sorbent tubes was designed as a proxy for ambient sampling conditions. Gaseous calibration standard aliquots were loaded to the sorption tubes in similar mass quantity as for liquid calibration standard aliquots. Sequences of measurements were carried out over a period of one month using gaseous calibration standards prepared at least every week and stored in between at 60°C (U.S. Environmental Protection Agency (EPA), 1999a). The raw data were adjusted for outliers using a 2-sided Grubbs outlier test (Grubbs, 1969) with a p-value criterion of < 0.05. Residual data were filtered according to a two-sigma (2 σ) criterion on the $\delta^{13}C$ and subsequently on the recovered compound masses. According to a Student-t-Test, means of filtered $\delta^{13}C$ values obtained from both calibration standard types originated from the same population. Still, means of gaseous calibration standards showed a minor enrichment in $^{13}C$ when compared to liquid standards (Table 1). Volatilization of CE is reported to show a minor $^{12}C$-enrichment in the residual phase with a magnitude similar to measurement uncertainties (Huang et al., 1999; Jeannottat and Hunkeler, 2012; Poulson and Drever, 1999). Therefore, differences in $\delta^{13}C$ mean values of recovered CE were assigned to handling issues of the gaseous calibration standards, which indicated higher standard

deviations when compared to the recovered CE obtained by the liquid calibration standard (Figure 2). Compound breakthrough due to the loading procedure was discarded, because it would have resulted in even more pronounced [13]C-depletion of the recovery CE (Klisch et al., 2012). Gaseous calibration standards were prepared and stored according to reported recommendations (U.S. Environmental Protection Agency (EPA), 1999a). A further decrease in the recommended maximum operation time to 48 h resulted in some improvement on the subsequently determined mass recovery rates. Plotting mass recovery rate versus the spiked amount of compounds, as shown in Figure 3, revealed that recovery rates were lower at smaller calibration standard aliquots. Nevertheless, poor recovery rates seemed to level-out above higher calibration standard aliquots of 2.2, 1.8, 1.3 and 1.3 nmol (PCE, TCE, cDCE and tDCE). The latter was assigned as the minimum quantification limit (MQL) for mixing ratios and stable carbon isotope ratios of CE for the presented measurement setup and represented a compound molar ratio of 105, 84, 64 and 63 nmol mol$^{-1}$, if sorbent tubes were loaded at a flow rate of 50 ml min$^{-1}$ over a sampling time of 10 minutes. Such MQL represents a sufficient sensitivity for ambient air monitoring applications (Hartwig, 2017; Maceira et al., 2017; Ras-Mallorquí et al., 2007; Woolfenden, 1997).

The relative standard deviation for mass recovery of each compound and calibration standard agreed with previous recommendations of less than 10% (Bianchi and Varney, 1993). Influences due to humidity were neglected, because (1) Tenax-filled tubes did not show an influence in the presence of humidity (Maceira et al., 2017) and (2) mass recovery from liquid standard aliquots showed more complete and reproducible mass recovery. Nevertheless, mass recovery rates suggested that using liquid calibrations standards was better than using gaseous ones. To conclude, using liquid calibration standards is preferred, because of smaller uncertainties of mass recovery and less fluctuation in $\delta^{13}C$ values.

**Figure 2: Comparison of $\delta^{13}C$ values obtained from sorption tubes loaded with either gaseous or dissolved calibration standard aliquots (nmol) for PCE, TCE, cDCE and tDCE. The dotted lines indicate the set values for $\delta^{13}C$.**

**Figure 3: Relative mass recovery rates of PCE, TCE, cDCE and tDCE when measured by the presented thermal desorption method after loading of gaseous or liquid calibration standard aliquots over the calibration range of chlorinated ethenes (nmol).**

### 3.2 Calibration of carbon dioxide and methane using glass vials

Precisions of $\delta^{13}C$ values (1 σ) of $CO_2$ and $CH_4$ were 0.13 ‰ and 0.23 ‰, respectively, when determined from working gas calibration standards (n=9, $CO_2$: 210 – 960 µmol mol$^{-1}$, $CH_4$: 550 – 2700 µmol mol$^{-1}$) and extended over the atmospheric background levels (for 2021: $CO_2$ at 413.2 µmol mol$^{-1}$ and $CH_4$ at 1889 µmol mol$^{-1}$ (WMO - World Meteorological Organization, 2021)). The precision in molar ratio from the same measurements (1 σ) was ± 2 µmol mol$^{-1}$ for $CO_2$ and ± 0.11 µmol mol$^{-1}$ for $CH_4$. Detailed information is provided in Leitner et al. (2020).

### 3.3 Field sampling campaigns

#### 3.3.1 Former domestic waste dump

##### 3.3.1.1 Whole air sampler

$CO_2$ molar ratios of whole air samples were found at 371 to 404 µmol mol$^{-1}$ (1 σ ≤ 5.6 µmol mol$^{-1}$) with $\delta^{13}C$ values of -10.4 ‰ to -9.2 ‰ (1 σ ≤ 0.21 ‰) as shown in Figure 4. For $CH_4$, molar ratios were found between 2.05 to 4.34 µmol mol$^{-1}$ with $\delta^{13}C$ values of -56.1 ‰ to 47.7 ‰. Each whole air sample was first analysed for $CO_2$, via three measurements of 300 µL sample gas aliquots, before being analysed for $CH_4$ using the residual whole air sample volume of approx. 20 mL.

**Figure 4: Results for $CO_2$ and $CH_4$ molar ratios and $\delta^{13}C$ values obtained from samples taken above an active (in operation) and non-active (in stand-by) landfill gas suction system unit. Error bar of $CO_2$ were obtained by measurement triplicates (n=3).**

Figure 4 illustrates the results for $CO_2$ and $CH_4$ measurements from samples taken above the active and non-active landfill gas extraction system. A Welsh two-sample t-test could not confirm that means of the molar ratio or the $\delta^{13}C$ values from active and non-active sampling spots were significantly different. Nevertheless, data points of the non-active pipeline incorporated three outliers (according to a Grubbs outlier test), which were indicated in Figure 4 as D6, D7, D14. It was hypothesized that those were biased by local emissions of $CO_2$ and $CH_4$ through the surface sealing originating from microbial degradation of organic waste

materials. Estimates of global $CH_4$ emissions rank waste disposals in the top-5 of anthropogenic methane sources. (Fowler et al., 2009) Therefore, a Keeling Plot (Keeling, 1958) of the latter three points was used as a tentative proxy to link the outliers to the formation of methane at the landfill. The estimated source signals were -20.2 ‰ (R-squared: 0.999) and -60.2 ‰ (R-squared: 0.718) for $CO_2$ and $CH_4$ respectively. Both source signal values clearly did not reflect the atmospheric background (annual means $CO_2$: -8.7 ‰ ± 0.5  (2015) and 419 ± 8 µmol mol$^{-1}$ (2021) at the NOAA Station Hegyhatsal, Hungary, which is the nearby NOAA station (White et al., 2015), and global annual mean $CH_4$: -47.3 ‰ and 1869 µmol mol$^{-1}$ (WMO - World Meteorological Organization, 2021)). Despite this preliminary finding a more precise interpretation would require the $\delta^2H$-$CH_4$ values to confirm, that the source values of the $\delta^{13}C$-$CH_4$ pointed towards formation of $CH_4$ due to microbial activity, which was indicated by depleted values when compared to the atmospheric background (Whiticar, 1999). Nevertheless, the $\delta^{13}C$ values for $CO_2$ and $CH_4$ in Figure 4 fell within the same characteristic range as previously shown for landfill gas emissions (Hackley et al., 1996). Methanogenesis from $CO_2$-$H_2$ was shown to yield a $\delta^{13}C$-$CH_4$ value of ~ -60 ‰ (Krzycki et al., 1987), while the pathway via $CO_2$ reduction was less likely because the $\delta^{13}C$-$CO_2$ source value would therefore have pointed to a more enriched value (~ -14‰) (Botz et al., 1996). Concomitant fermentation of the organic waste to supply the metabolic need of dissolved organic carbon and $CO_2$ respectively would need a $\delta^{13}C$-value of ~ -22 ‰, which is close to previous observations (Mohammadzadeh and Clark, 2008). Although the former landfill is equipped with a landfill gas extraction system and surface sealing, minor landfill gas leakage of $CH_4$ could be identified due to the incorporated fraction of the locally emitted biogenic footprint of $CH_4$, as also shown in previous studies (Bakkaloglu et al., 2021).

### 3.3.1.2   Sorption tube sampler

Analysis of sorption tubes did not reveal any local emissions of CE. However, heptane and toluene could be detected in one or two sorbent tubes taken above the ex-situ filter facility. The molar ratio of heptane and toluene was found at 20 nmol mol$^{-1}$ and 30 nmol mol$^{-1}$, respectively. Compounds were identified according to their MS spectra (Wallace, 2022) and verified and quantified by measurement of gaseous calibration standard aliquots (~1000 µmol mol$^{-1}$) of 20 to 50 µL, adding heptane and toluene before loading them to sorbent tubes similar as for CE. The measurement setup was also similar as for the CE. $\delta^{13}C$ values of the compounds used in the calibration standard and obtained by field-derived samples agreed by less than 0.5 ‰ (-27 ‰ for toluene, -29 ‰ for heptane), which was an indication that both chemicals were once produced from similar resources like coal tar or crude oil.

### 3.3.2   Forest demonstration center

Results showed $CO_2$ molar ratios over a narrow range of 382 to 404 µmol mol$^{-1}$ with σ ≤ 7.3 µmol mol$^{-1}$ and $\delta^{13}C$ values of -7.5 ‰ to – 8.8 ‰ with σ ≤ 0.3‰ (n=9). Values were obtained by successive analysis of three 300 µL whole-air sample aliquots from each sample vial ("measurement triplicates"), which were mostly sampled in triplicates at each sampling height and location during the field sampling campaign. Measurement triplicates of individual whole-air samples showed a σ ≤ 6.6 µmol mol$^{-1}$ and ≤ 0.2 ‰ with no significant difference in σ between manually and UAV-collected samples.

**Figure 5: Comparison of $CO_2$ molar ratios and $\delta^{13}C$ values obtained by UAV-based (black) and manual sampling (white, red) at the forest demonstration site. Dotted lines indicate consecutive sampling. Grey rectangles represent the sampling height (pointes were dogged to increase visibility of individual samples). Error bars were obtained from measurement triplicates (n=3).**

Figure 5 illustrates the vertical profiles for molar ratios and $\delta^{13}C$ values of $CO_2$ obtained from manual and UAV-based sampling, indicating the consecutive sampling path of each group by the dotted lines. Initial manual sampling at the tower ("manual at tower") was done around noon, starting approx. two hours prior to the UAV-based sampling procedure, which also included some manual sampling. The latter comprised of simultaneous manual ("manual at UAV") and UAV-based air sampling at a height of 0.4 m with the UAV-rotors not yet turned on ("UAV rotors off") and the UAV-based sampling with rotors turn on along the ascent of the

vertical profile ("UAV rotors turned on"). With the exception of the final manual sample at 0.4 m during the descent of the tower, all samples were collected in triplicate.

The largest variation in $CO_2$ molar ratios was observed for the sampling height of 0.4 m. Results showed a higher molar ratio of $CO_2$ for the final manual sample at 0.4 m at the tower (indicated by the red triangle in Figure 5) compared to the initial three samples, but similar values of $\delta^{13}C$, which is why the increase in the molar ratio was reasoned by slightly elevated air temperatures (sample gas density) compared to the start of the manual sampling. The latter was also argued for the slight offset of the manual and UAV-assisted vertical profile data. The variation of $CO_2$ molar ratios between simultaneous manual sampling at the UAV and sampling by the UAV itself could have been caused by the breath of the operators while walking around the UAV and/or by the operators walking over and disturbing forest floor vegetation and the moist soil layer. In conclusions, fluctuations in molar ratios and $\delta^{13}C$ values, were attributed to the diurnal variability in ecosystem respiration (Ehleringer and Cook, 1998), which for the presented data covered a range of $< 30 \, \mu mol \, mol^{-1}$ and 2.5‰ and seemed to level out with increasing sample height. Factors, such as photosynthesis and soil respiration, usually maintain the $\delta^{13}C$ source signal below -22‰ (Cernusak et al., 2013; Hemming et al., 2005). The expected pattern of a vertical profile at the sample location, starting from the ground level, must therefore follow continuous $^{13}C$-$CO_2$-enrichment when approaching the atmospheric background level of $CO_2$ ($\delta^{13}C \sim$ -8.5‰ (Rubino et al., 2019)). The diurnal variation in the $CO_2$ molar ratio has been shown to be more pronounced closer to the ground level and that diurnal changes at ground level are of a higher magnitude than variations along vertical profiles from forest floor to above the canopy (Bowling et al., 2005; Buchmann et al., 1998).

The sampling entrance point was located at the centre and above of the UAV-rotor plane, which was reported as the location with the minimal impact due to the rotation of the rotors (Zhou et al., 2018). It was assumed that the influence of the UAV rotors, due to the downwash generated by the propellers and the resulting disturbed air flow field, was the factor most influencing the vertical falsification of the actual sampling point, especially when the UAV operated close to ground level (Burgués and Marco, 2020; Zhou et al., 2018). UAV-assisted sampling took place during the ascent with a dwell time at a constant altitude of about 1.5 minutes, which is needed to take three sample replicates. According to Brosy et al. (2017) sampling during the ascent ensures that the air is not mixed by the UAV before sampling is initiated. Andersen et al.( 2018) compared UAV-based whole air samples taken during the ascent and descent and relative to a tower. The reported variation was $< 13 \, \mu mol \, mol^{-1}$ comparing UAV-based samples and tower measurements and standard deviations were similar for the ascent and for the descent. Observing results from manual sampling at the UAV and UAV-based sampling with rotors on (Figure 5) showed that $\delta^{13}C$ values of manual sampling pointed towards the impact of soil respiration while UAV-based samples indicated continuous $\delta^{13}C$-enrichment of residual $CO_2$, which was linked to photosynthetic activity of the overlying layers pursued by the consecutive UAV-based samples at the height of 5 and 13 meters. The impact of downwash seemed to level-out with the last sample of height 13 m, thereafter showing similar $\delta^{13}C$ values from manual and UAV-based sampling. The apparent offset in molar ratios between manual and UAV-based samples was attributed to the difference in sample time as already pointed out for the example of manual sampling at the tower at the height of 0.4 m with 120 minutes in between.

## 4    Conclusions

The most demanding step, as for most analytical systems, is the implementation and testing of the initial workflow of the sampling procedure. Here we present a sampling system coupled to an off-line measurement setup to measure atmospheric $CO_2$, $CH_4$, and VOC molar ratios and $\delta^{13}C$-values at ambient conditions. Two samplers and a whole-air sample vessel preparation device were developed and evaluated in the laboratory and at field conditions, while the measurement setup was evaluated in prior experiments (Leitner et al., 2020). That measurement setup enabled the determination of the mole fraction and stable isotope ratio of carbon of the target compounds detailed herein and could be further applied to investigate the stable isotope ratios of hydrogen in $CH_4$, oxygen in $CO_2$ as well as oxygen and nitrogen in nitrous oxide ($N_2O$).

The samplers can be easily mounted to any unmanned aerial vehicle with sufficient payload capacity (~ 1 kg), making it simple to sample at remote places or conduct automated sampling missions. The sampling system was tested at two field sites. A comparison

with manual sampling revealed reasonable compatibility with the UAV-based sampling method. The results also showed that the system sensitivity is sufficient to detect $CO_2$ and $CH_4$ emissions and stable isotope signatures close to atmospheric background molar ratios, for which otherwise extensive and expensive sampling flights are required (Bayat et al., 2017), thereby providing an alternative to traditional approaches (Mønster et al., 2019).

Although we have proven the functionality of this system, location-specific sampling especially for vertical sampling profiles close to ground level, due to the impact of UAV-rotor downwash, needs further investigation. Such impact to the air field surrounding the UAV is thought to be dependent on the applied UAV-specifications (Shukla and Komerath, 2018), thereby limiting the scope for general recommendations.

## 5    Acknowledgements

The authors acknowledge financial support by the Austrian Research Promotion Agency (FFG) within grant 866949 "UAV-based gas monitoring systems for the underpinning of urban, agricultural and industrial emission roadmaps" of the Beyond Europe 2nd Call

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

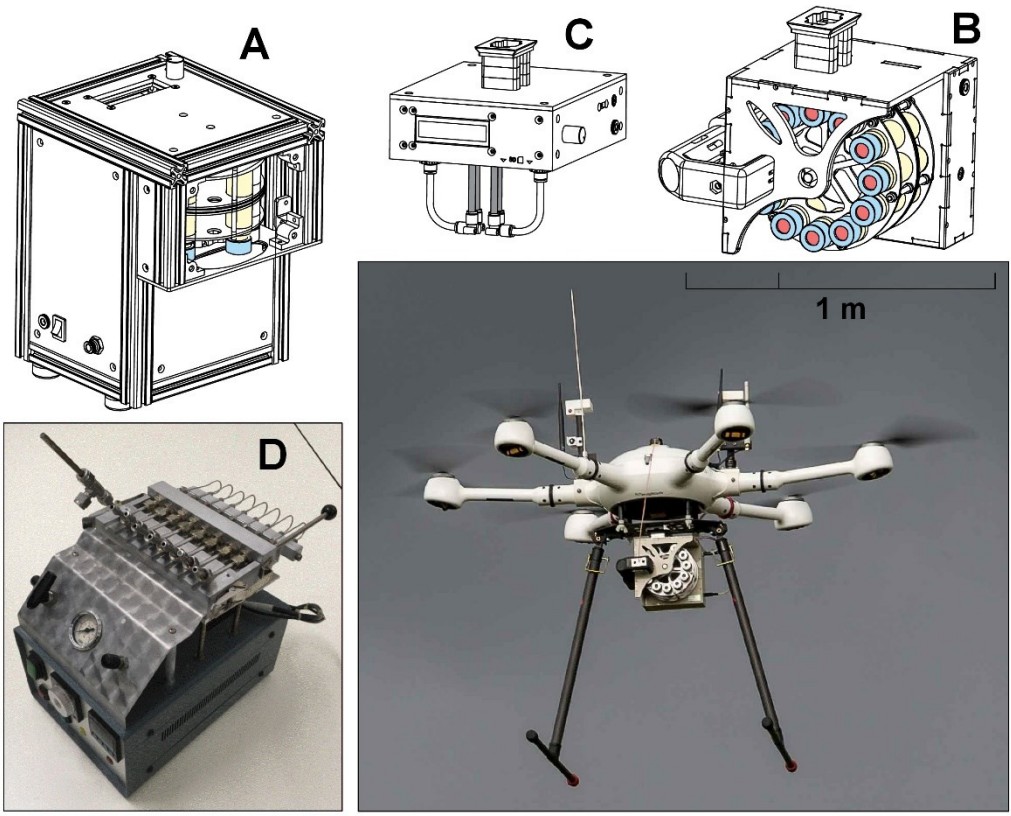

**Figure 1: Overview of the sampling systems comprising of the vial preparation device (A), the whole-air sampler (B), the gas sampler for**
**adsorbent tubes (C), the sorbent tube conditioning device (TubeCon2, D) with one sorbent tube attached and the UAV applied in field tests equipped with the whole-air sampler.**

**Table 1. Comparison of $\delta^{13}C$ mean values ± twice the standard deviation (2 σ) and mass recovery rates, as means ± 2 σ, obtained from measurements of gaseous and liquid calibration standard aliquots at the given mass range (nmol) of chlorinated ethenes (PCE, TCE,**
**cDCE, tDCE) loaded to sorbent tubes.**

| calibration standard | compound | $\delta^{13}C_{mean}$ ± 2 σ | mass recovery (mean ± 2 σ) | nmol (min – max) | n |
|---|---|---|---|---|---|
| gaseous | PCE | -27.3 ± 0.5 | 0.63 ± 0.22 | 2.2 - 4.4 | 34 |
| liquid | PCE | -27.5 ± 0.1 | 1.00 ± 0.06 | 2.1 - 3.5 | 12 |
| gaseous | TCE | -29.5 ± 0.4 | 0.81 ± 0.17 | 1.8 - 4.4 | 29 |
| liquid | TCE | -29.8 ± 0.1 | 1.00 ± 0.05 | 1.6 - 4.1 | 31 |
| gaseous | cDCE | -25.9 ± 0.7 | 0.82 ± 0.11 | 1.3 - 4.4 | 39 |

| liquid | cDCE | -26.0 ± 0.08 | 1.00 ± 0.04 | 1.3 - 4.4 | 26 |
|--------|------|--------------|-------------|-----------|----|
| gaseous | tDCE | -12.2 ± 0.4 | 0.77 ± 0.13 | 1.3 - 4.4 | 43 |
| liquid | tDCE | -12.2 ± 0.05 | 1.00 ± 0.03 | 1.6 - 3.9 | 32 |

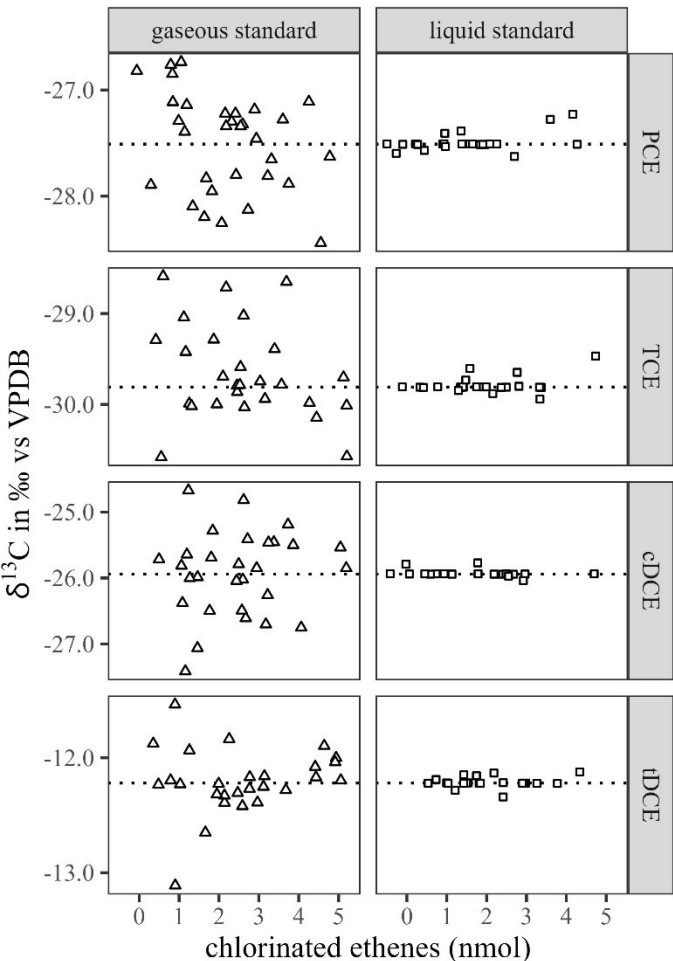

**Figure 2: Comparison of δ¹³C values obtained from sorption tubes loaded with either gaseous or dissolved calibration standard aliquots (nmol) for PCE, TCE, cDCE and tDCE. The dotted lines indicate the set values for δ¹³C.**

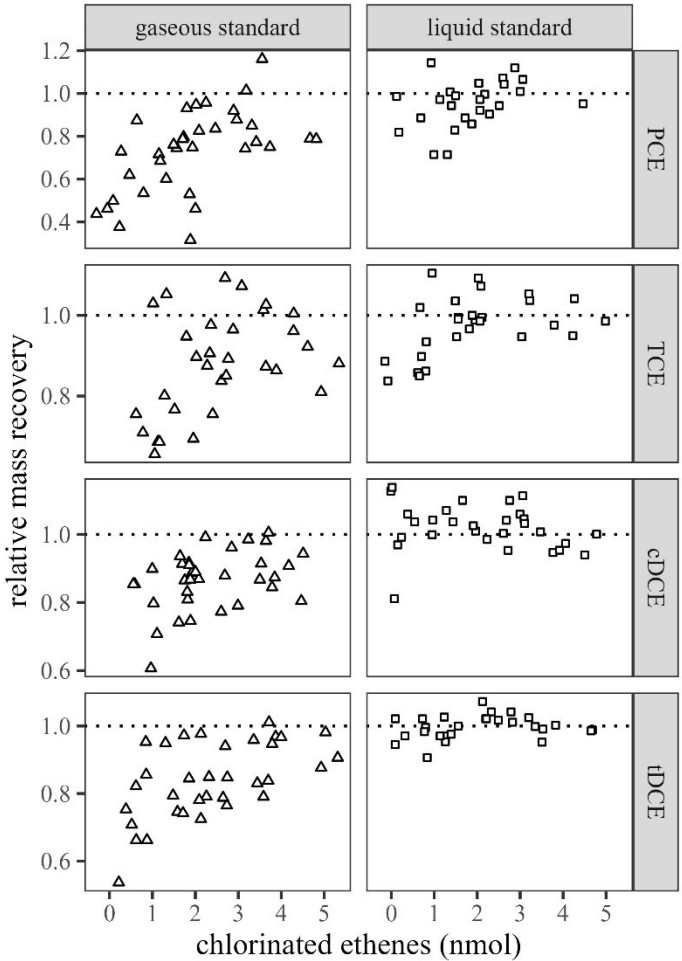


**Figure 3: Relative mass recovery rates of PCE, TCE, cDCE and tDCE when measured by the presented thermal desorption method after loading of gaseous or liquid calibration standard aliquots over the calibration range of chlorinated ethenes (nmol).**

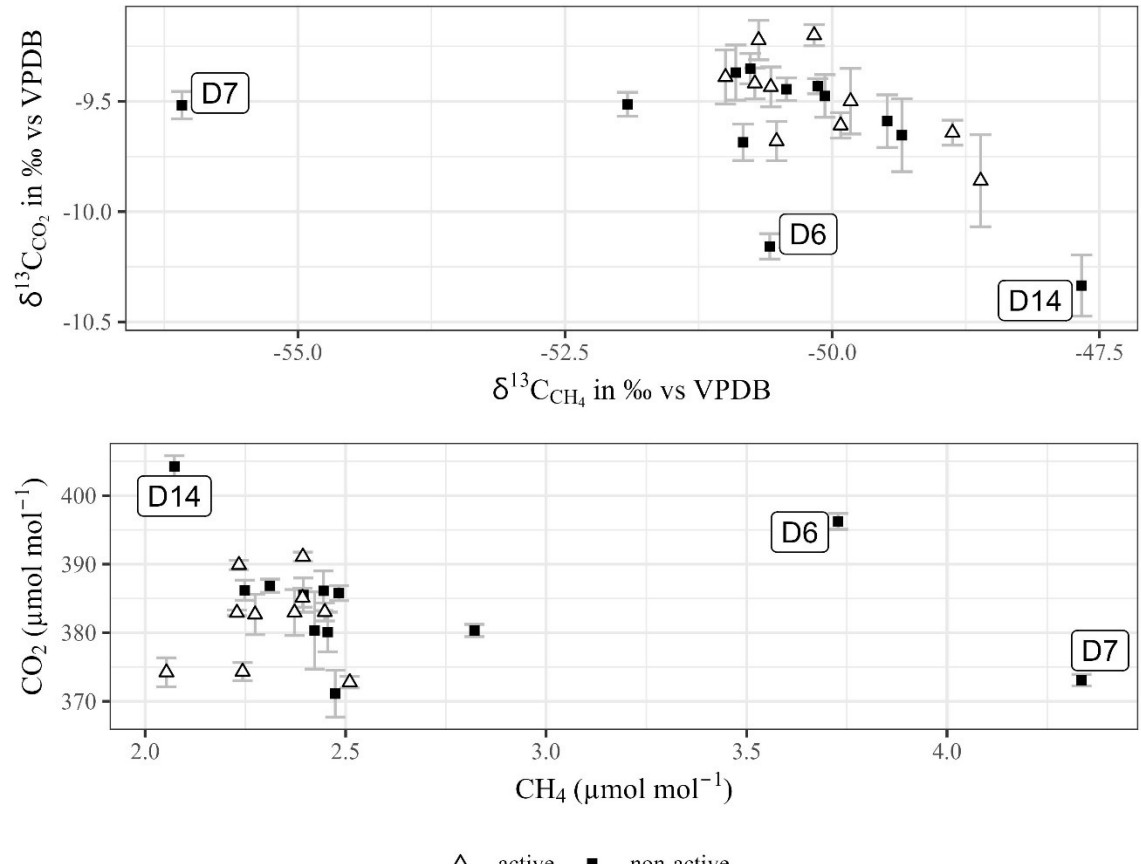

**Figure 4: Results for CO₂ and CH₄ molar ratios and δ¹³C values obtained from samples taken above an active (in operation) and non-active (in stand-by) landfill gas suction system unit. Error bar of CO₂ were obtained by measurement triplicates (n=3).**

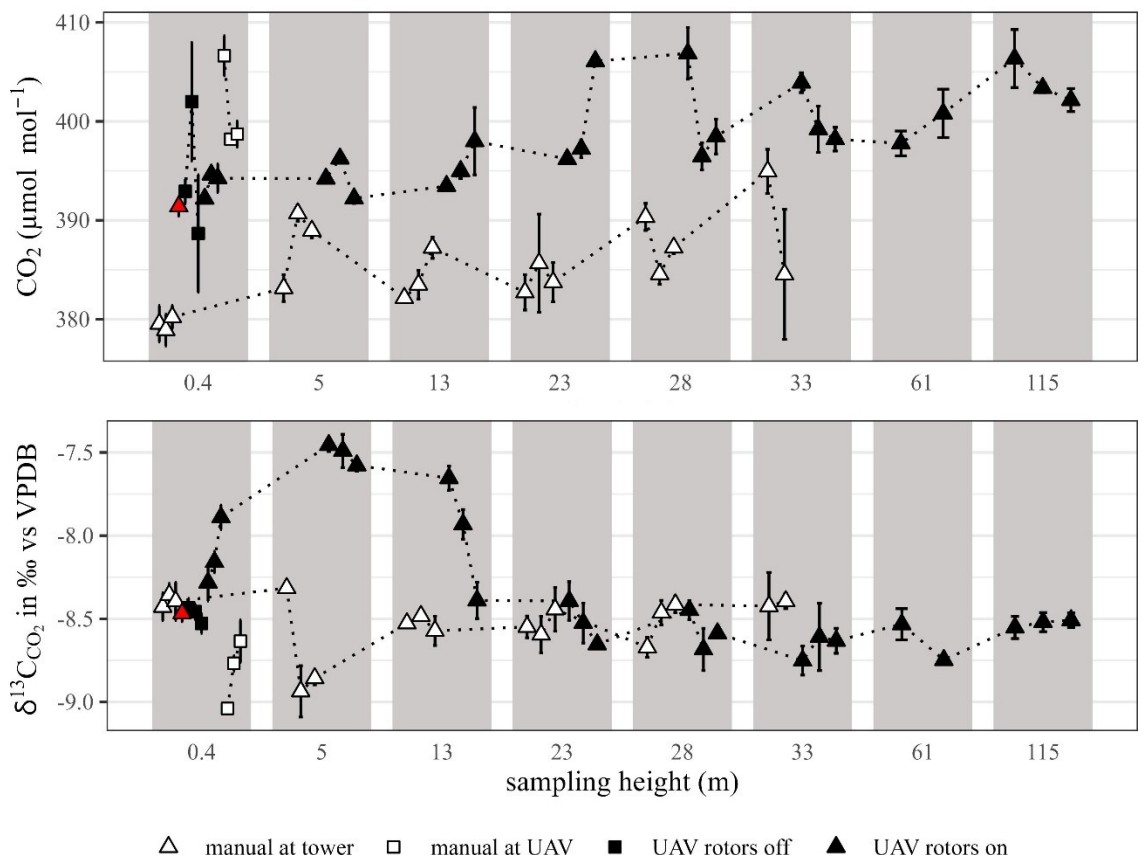

**Figure 5: Comparison of CO₂ molar ratios and δ¹³C values obtained by UAV-based (black) and manual sampling (white, red) at the forest**
**demonstration site. Dotted lines indicate consecutive sampling. Grey rectangles represent the sampling height (pointes were dogged to increase visibility of individual samples). Error bars were obtained from measurement triplicates (n=3).**