# Peer review of "UAV-based sampling systems to analyse greenhouse gases and volatile organic compounds encompassing compound specific stable isotope analysis"

_EGUsphere, 2022_

## Author Comment (AC1)

**Summary:**

This study describes the characterisation and operational use of an ambient gas sampling system consisting of sorbent tubes and glass vials installed on a UAV with MTOW < 25 kg. Target gases include CO2, CH4 (and their C-13 fractionation), and a range of VOCs, all measured offline (in the lab) by GC-C-IRMS etc. Performance is described with reference to certified gas and C-13 isotope standards and interpreted for mass recovery. Finally, some interesting field measurements are reported for these gases by UAV sampling near to the surface over both a landfill and forest site.

This review is limited to ambient atmospheric sampling, UAV operations, C-13 isotope measurements and GHG concentrations in general, as I am not a specialist in offline GC-IRMS methods or sorbent tube sampling.

In terms of context, this paper is most welcome as it continues to push the boundaries of what UAVs can offer as an atmospheric sampling platform. A system that can use UAVs to take whole air samples for later high-precision laboratory analysis is an interesting and novel contribution to a rapidly growing measurement field. Measurements such as those described are likely to have high scientific value to others in the field of GHG and VOC emission research.

Overall, the paper seems to have excellent scientific value and originality and high standards of academic rigour. However, the presentation of the paper does not do the work justice. It is currently very hard to read and absorb and contains many grammar mistakes and sentences that make very little sense. However, this could be fixed with a thorough proof-read and edit by an experienced technical writer. Some technical aspects are not described as completely as might be necessary for this to be an excellent resources for others to follow. It would be a much better paper if it can be written a little better.

I have a technical question about the nature of the WAS system and how this may impact the accuracy of ambient measurements of GHGs and their isotopic fractionation (see below). My main comment is that the paper needs revision to make it more accessible - a relatively minor correction, and some attention to detail on figure captions etc.

*We thank you for your helpful review and can present you with a completely revised version. The content of the revised manuscript was corrected by a scientific colleague with the same topic and who is native English speaker in order to hopefully be able to present you with a comprehensible manuscript*

**Specific comments:**

1/ Line 11  - In the abstract it is stated that the instruments can be mounted to UAVs (and reads/implies that any UAV can carry them). However, not every UAV can carry them (some UAVs are extremely small and can carry no payload). I would suggest that this sentence is better qualified by saying "…can be mounted to a UAV with a payload capability greater than X kg…" and insert what x should be. Moreover, this sentence overall does not make sense grammatically – insert a full stop before "for all compounds…".

*Revised*

2/ Line 20 – what are "replicates" in this context? I've not heard this terminology before. Also, what is meant by "triplicate" CO2 measurements – does this mean that the same sample was measured three times to arrive at the stated precision, or that three measurements of different samples were used to calculate the precisions? I'm struggling to understand what was done to arrive at the precisions stated in the abstract here as the terminology is very confusion.

*The abstract was modified to avoid such awkward wording and provide the reader with a clearer overview. The meaning of "replicates" and "triplicates" is now explained more clearly in the methods and results.*

3/ Introduction. The literature review may be well served by citing and discussing a recent review paper of UAV methane measurements by Shaw et al., 2021 - https://royalsocietypublishing.org/doi/full/10.1098/rsta.2020.0450

*The paper of Shaw was added to the introduction, because it provides a helpful summary of available sampling and measurement setups including payloads and measurement precision.*

4/ Line 70 – the studies cited here are examples of such studies and not an exhaustive list as the list of citations implies. If you only wish to cite these three examples, please add "e.g.". Otherwise, a more complete list of UAV GHG measurement studies up to 2021 is described in Shaw et al., 2021 review paper (link in the comment above).

*Revised.*

5/ Section 2.2.2. The WAS system appears to consist of 12 evacuated (flushed and pre-treated) glass vials. Injection of ambient air appears to be via a side-port needle. Are these vials evacuated to 0.5 Pa as suggested on line 151? If not under high vacuum, the injected (ambient) air would be mixed with whatever residual air was in the vial, potentially biasing the concentration and C-13 fractionation measured offline (by high precision instrumentation in the lab) toward the concentration and fractionation of the residual air inside the vial. The final sentence of this section states that "The dead-volume of the transfer line (100 μL) and the residual flush-gas volume inside the evacuated glass vial did not show any significant influence to follow-up measurement setup." This is an example of a sentence that doesn't make complete sense to me (i.e. what is "follow-up measurement setup?). Please could you provide more information/data on why you are confident that there was no significant influence, and what you define "significant influence" to mean (quantitatively)? I am concerned that this effect could greatly bias the GHG concentration and isotope measurements made from the filled vials. GHG concentration measurements are required to have a very high precision to be acceptable under WMO reporting standards. The text reads as though it is only the initial negative pressure that fills the tube to an equilibrium pressure with ambient pressure. If the vials are indeed evacuated to 0.5 Pa (high vacuum), and there is only 100 ul residual in the transfer line, I would agree any bias would be very small, but it may be useful to quantify the maximum expected bias this may introduce for each quantity measured. Please could you explain and clarify this in the response and in the manuscript.

*Revised. Vials were evacuated using a rotary vane pump, which can achieve 0.5 Pa. Beforehand, vials were flushed with synthetic air to ensure that the residual gas volume after*

*the evacuation does not contain residual air constituents. The description of the vial conditioning now also incorporates the results of conditioned vial, which were then filled with synthetic air to reveal measurement blanks.*

6/ Figure 2 – the caption does not describe what the error bars represent – what do these represent?

*Revised*

7/ I'm pleased to see the honest discussion about the potential effect of propeller downwash, especially for near-surface measurements where disturbed air may be sampled, making it very hard to assign a measurement height to sampled air. Others have tackled this by only recording measurements on rapid descent profiles (or rapid ascent profiles where vertical profiles are required), or by taking measurements forward in the direction of motion horizontally (moving fast enough such that downwash does not have time to mis/disturb the air ahead of the UAV). It may be useful to discuss this as a potential mitigation to these sampling problems.

*Revised. References in concern of flight and sampling paths were added.*

**Technical comments:**

Line 15: Change to "…sampling or collected and sampled…"

*Revised*

Line 23: "The  results emphasized  the functionality
of  the  sampling  and  measurement  setup described, demonstrating that it a viable tool for monitoring atmospheric trace gas inventories and identifying emission sources."  - This is a bit superfluous and confusing – recommend changing text to read: "The results demonstrate that the UAV sampling system here represents a viable tool for monitoring atmospheric trace gas inventories and identifying emission sources". This is just one example of many sentences that are rather verbose and partially meaningless. It may also be useful to emphasise the isotopic capability of this system versus other in situ measurements of trace gas cocentrations, and how the two may complement each other (a combination of such systems could be very powerful).

*Revised*

There seems to be an inappropriate use of commas in many places throughout the manuscript, e.g. the sentence beginning on Line 45 does not need a comma.  Please proof read and check correct use of commas throughout.

*Revised*

Line 50: This sentence makes no sense. I think a full stop may be missing between "precision" and "sample"?

*Revised*

Line 391: What does "chilled" mean in this sentence? Was the sampling entrance point chilled?

*Revised*

Line 395: "…of height 13,…" please add "m" after "13".

*Revised*

**Citation**: https://doi.org/10.5194/egusphere-2022-830-RC1

---

## Author Comment (AC2)

Review of the paper

A UAV-based sampling system to analyse greenhouse gases and
volatile organic carbons encompassing compound specific stable isotope analysis

by Leitner et al.

This paper describes the development of air samplers for the analysis of $CO_2$ and methane as well as some selected VOC, in this case light chlorinated hydrocarbons. The samplers can be mounted on a small UAV. The samples are then analyzed in the laboratory. In general, the approach is very interesting because such unmanned aerial vehicles can be used where the collection of air samples is difficult to perform, e.g., directly at certain sources such as smokestacks, volcanoes, or at low altitudes over rough terrain. Therefore, the development of such methods is very welcome. This work certainly makes an important contribution to this field and is of great interest to the scientific community. The functionality of the system is proven at the end by some measurements, mainly mixing ratios and isotope ratios of $CO_2$ and $CH_4$. The results of the VOC measurements, on the other hand, are less convincing.

In principle, the paper is worth publishing, but not in its present form. It needs to be completely revised linguistically. Also some explanations and descriptions should be formulated more clearly.

General comments:

The paper is not easy to read, firstly because the English is not particularly good, and secondly because many sentences are completely incomprehensible. Some sentences have to be read several times to understand what is meant. Often enough, the sentences are grammatically incorrect and lack periods and commas.

Examples are the sentences on page 1, line 11: "Both samplers can be mounted to an unmanned aerial vehicle (UAV), the targeted compounds were greenhouse gases (e.g. $CO_2$, $CH_4$) and volatile organic compounds (VOC, i.e. chlorinated ethenes), for all compounds mole fraction and the stable carbon isotope ratio were measured." or on page 2, line 50: "However, sampling methodology is attendant on the target measurement precision sample pre-requisites for GHG measurements are similar to those of VOCs, often relying on large and heavy sample containers (International Atomic Energy Agency, 2002)."

Cryptic sentences can be found throughout the paper. Numerous imprecise statements considerably reduce the value of this paper. A thorough proofreading and a substantial revision are necessary here.

*We thank you for your review, as it provided valuable comments for improving the manuscript. The revised manuscript was linguistically and grammatically corrected by a scientific colleague who is a native English speaker.*

In the introduction, a lot of information is given in a general way. However, some explicitly refer to the measurement of $CO_2$ with high mixing ratios, while others refer to measurements of VOCs with extremely low mixing ratios. A clearer distinction should be made here, since the prerequisites are completely different, especially for the isotope measurements.

*It should be noted at the outset that an error crept into the units for greenhouse gases. In many places mmol mol-1 was given instead of µmol mol-1. Now it can be seen that both the greenhouse gases and the volatile organic compounds were present in similar concentration ranges and were analysed from substance quantities of a few nmol.*

*We hope we have correctly interpreted your comment about the different mixing ratios, that this was provoked by the wrong units.*

*The introduction was revised to deal not only with carbon dioxide but also methane. Furthermore, references were added.*

Why was the focus set to CE, which are maybe not the most important VOC as suggested?

*CE are often found at former landfills and were therefore of interest to us. Since we also work with chlorinated ethenes in the field of groundwater contamination, we already have relevant knowledge in the analysis of these substances from aqueous samples, which we were able to use in the selection and evaluation of adsorbent materials and in the determination of the recovery rates of the CE.*

In the title, "volatile organic carbons" should be changed to "volatile organic compounds".

*Revised*

Specific comments

Page 1, line 19: nmol should read nmol $mol^{-1}$

*The Abstract was revised. nmol was the right unit, but it was meant to be "molar amount range", not "molar ratio range".*

Page 1, line 20: What is s triplicate measurement of a replicate? Do you really mean 7.3 mmol $mol^{-1}$? To be able to assess the data, it would be necessary to know how large the sample volume was. Was the sample partitioned and if so, why? Was each sample measured three times? There are more questions than answers here.

*Revised and formulated more generally.*

*The definition of the terms "triplicate" and "replicate" is now included in the methods and results. In addition, the sample volumes and the aliquot gas volumes for the measurement have now been explained more clearly in the methods and results.*

*μmol mol-1*

Page 2, line 41: Reference is made here to the additional information that measurements of the ratios of stable (carbon) isotopes can provide. Works by Keeling (1958 and 1979) are cited, which refer exclusively to $CO_2$. There is a lot of more recent literature on isotope ratios of $CO_2$, but also especially of VOC. Current and especially specific literature should be cited here.

*This section has been supplemented with references. We want to point out that the introduction is specifically designed for the experience with UAV-based sampling systems and the connection between sampling system and analysis in the laboratory, which also allows for stable isotope analysis. Therefore, and due to the fact that we wanted to keep the introduction manageable, we hope that the current version meets your expectations.*

Page 3, line 80: I think local sampling provides information about the emissions at the time of sampling, but not about inventories.

*Revised*

Page 3, line 93 ff.: Did you test the criteria that led to the selection of Tenax GR yourself or does the selection refer to the cited literature? What problems were encountered with the other adsorbents? And what was the recovery rate? Which ghost peaks occurred?

*This section has been revised. Tenax was chosen because it was the best option with respect to our analytical measurement setup (the desorption temperature of the autosampler is limited to 280°C, which is suboptimal for the use of strong adsorbent materials such as carboxen). We tested all listed adsorbents with our measurement setup, but selected the tested adsorbents in advance in accordance with the literature. The term "ghost peak" refers to the wording of the cited paper in which they saw peaks of unknown compounds that might have resulted from the decomposition of the adsorbent material, if I remember correctly. We also saw such peaks in our own chromatograms. These peaks also interfered with some of the CE, making those adsorbent materials unsuitable and hindered the calculation of recovery rates according to the sampling procedure presented. We do not think that I is of great interest for the audience to present those pre-experiments, which is why we focused on the results of Tenax only.*

Page 5, line 131: PBT must be PET.

*Revised*

Page 5, line 151: "We recommend … ". This sentence makes no sense.

*Revised*

Page 5, line 155: Delete "developed"

*Revised*

Page 6, line 161: What is a "negative pressure"?

*Revised*

Page 6, line 162 ff.: "The dead-volume …" Did you measure the influence on the results? What does "significant" mean here? How large is the influence? What do you mean by "follow-up measurement set-up"?

*Revised.*

*The influence on the results is now explained and an MQL is given. "Follow-up measurement set-up" means that a sample vial is first placed in an auto sampler, which withdraws 300 μl of sample volume and analyses this volume to determine the molar ratio of carbon dioxide. This procedure is repeated three times with a fresh sample volume aliquot of 300 μl. Then the sample vial is placed in a second auto sampler on the same GC-IRMS system, which purges the vial to analyse the residual volume (20 ml - 3\*0.3ml) and determine the molar ratio of methane. This procedure had been presented in a previous publication of our group, which is cited in the manuscript.*

Page 6, line 183: Here a sampling time for the adsorption tubes of 600 s is given. With a flow rate of 50 mL/min as given in chapter 2.1.3 an air volume of 500 mL is sampled. As an example, assuming a DCE mixing ratio of some 100 pptV, this would result in some 10 ng of the compound and only a few ng of carbon per sample. Usually about 50 ng of carbon are needed to measure isotope ratios with a GC-C-IRMS with a suitable precision. I am not sure, if this method is well suited to measure isotope ratios of VOC.

*Similar to the other CE, cDCE was analysed from ~1 to 5 nmol cDCE, giving 2-10 nmol C, which is sufficient for continuous flow CSIA. The time of 600 s at a flow rate of 50 ml min-1 was only an example and was used to load the sorbent tubes with the calibration standards during the laboratory experiments. For use in the field, the sampling time can be increased, but care should be taken that the flow rates and sampling time, according to the manufacturers' specifications and the available literature, do not allow a large margin.*

Page 7, line 211: "electronic ionization" must be "electron ionization".

*Revised*

Page 7, line 239: "CH$_4$, which was ….". This sentence is another example of imprecise wording. Methane is not "oxidized" to H$_2$, but pyrolized. Did you really measure the isotope ratio of $^2$H/H? If not, this information is unnecessary here.

*Revised and H2 was excluded.*

*The measurement setup presented can be used to determine dH2-CH4, but we did not analyze our field samples for dH2-CH4 because we would have needed additional samples. One sample each would have been required for D13C-CH4 and dh2-CH4, and we took only one sample from each sampling point in the field.*

Page 9, line 273 ff.: "vegetation crown", better: "canopy height". And I think, not the field campaign was overcast, right? What do you mean by "decoupling from atmospheric background"? CO$_2$ profiles are not generated, but measured. With the described system mixing ratios or isotope ratios can be measured, but no fluxes.

*Revised*

Page 10, line 29: "Injected amounts …". Again a cryptic sentence.

*Revised*

Page 10, line 303: "Volatilization of light isotopes …" I think you mean "volatilization of molecules containing no $^{13}$C atoms …"

*Revised*

Page 10, line 315: The minimum quantification level is quite different for the measurement of mixing ratios and isotope ratios. The "sufficient sensitivity" may hold for ambient air monitoring but not necessarily for isotope ratio measurements. Maybe, a more specific discussion is needed here.

*Revised. The given MQL was valid for both, measurement of mixing ratios and isotope ratios.*

Page 12, line 363 ff.: The mixing ratios of heptane and toluene seem to be rather high. If these values are correct, I wonder why they are only given as approximate values. Since a GC-IRMS system was used (by the way, there is no information about the chromatography such as columns, temperature program, etc.), I wonder why only the two compounds were measured. With the comparatively high mixing ratios, I would expect that a large number of

other compounds should be measurable. Are the isotope ratios given for the calibration standard or for the ambient measurements?

*The method was already presented in chapter 2.5 - Measurement setup for sorbent tubes. The mixing ratios of heptane and toluene were indeed quite high, but could be justified with the landfill gas measurements listed in the references in Chapter 2.7 - Description of field sites. BTEX were detected in the landfill gas with a mixing ratio of ~10 μmol mol-1, which is ~1000 times the mixing ratio we determined with our sorbent tubes. In addition, we rechecked and evaluated measurements on the mixing ratios of heptane and toluene and arrived at the values as given. Heptane and toluene were the only compounds, which could be detected during sorbent tube analysis (according to a library search) and verified by preparing calibration standards with heptane and toluene to verify the results of the library search results. We have also revised the corresponding chapters to make them more understandable for the audience.*

Page 13, line 380 ff.: In this section it is not clear to me when exactly which sample was taken. First, it states that the manual samples were taken two hours before the UAV samples. Then it states the samples were taken at the launch of the UAV along with the UAV samples. What is meant by "chilled center"? What is meant by "rotor off"? My understanding is that the UAV would crash if the rotors were turned off. How were the samples collected with the UAV, in the ascent or in the descent phase? That has a significant impact on the air samples collected, as is implied in the text. I think a clearer description of the sampling process is needed here.

*Revised*

Figure caption, Fig. 1: What is A, what is C? There's not really anything discernible at D.

*Revised*

Figure 2: If you look at the results of the gas standards, there are data points with large error bars of almost 1 per mil, which I think is realistic, but also data points with practically no errors, which is quite unrealistic. Here I lack a more detailed explanation.

*Revised*

Figure 5: I suggest to show here an actual "vertical profile", namely the height on the y-axis. Beyond that, the axis label is quite odd.

*The figure has been revised, but we have kept the height on the x-axis because we think it is easier to read that way with error bars on the y-axis.*

**Citation**: https://doi.org/10.5194/egusphere-2022-830-RC2

---

## Referee Report (RR1)

Review of Leitner et al (revised)

Summary:

The revised manuscript satisfactorily addresses all comments raised in my earlier review. The quality of presentation (especially the clarity of the text and overall narrative) is much improved and the authors have made a substantial and appropriate revision. I have no further comments and recommend publication.